# Differential Effects of Cytokine Versus Hypoxic Preconditioning of Human Mesenchymal Stromal Cells in Pulmonary Sepsis Induced by Antimicrobial-Resistant *Klebsiella pneumoniae*

**DOI:** 10.3390/ph16020149

**Published:** 2023-01-19

**Authors:** Declan Byrnes, Claire H. Masterson, Jack Brady, Senthilkumar Alagesan, Hector E. Gonzalez, Sean D. McCarthy, Juan Fandiño, Daniel P. O’Toole, John G. Laffey

**Affiliations:** 1Anaesthesia, School of Medicine, University of Galway, H91 TK33 Galway, Ireland; 2Regenerative Medicine Institute (REMEDI) at CÚRAM Centre for Research in Medical Devices, Biomedical Sciences Building, University of Galway, H91 TK33 Galway, Ireland; 3Department of Anaesthesia, Galway University Hospitals, SAOLTA University Health Group, H91 YR71 Galway, Ireland

**Keywords:** MSCs, sepsis, pneumonia, pre-activation

## Abstract

**Background**: Pulmonary sepsis is a leading cause of hospital mortality, and sepses arising from antimicrobial-resistant (AMR) bacterial strains are particularly difficult to treat. Here we investigated the potential of mesenchymal stromal cells (MSCs) to combat established *Klebsiella pneumoniae* pneumosepsis and further evaluated MSC preconditioning and pre-activation methods. **Methods**: The potential for naïve and preconditioned MSCs to enhance wound healing, reduce inflammation, preserve metabolic activity, and enhance bacterial killing was assessed in vitro. Rats were subjected to intratracheal *K. pneumoniae* followed by the intravenous administration of MSCs. Physiological indices, blood, bronchoalveolar lavage (BAL), and tissues were obtained 72 h later. **Results**: In vitro assays confirmed that preconditioning enhances MSC function, accelerating pulmonary epithelial wound closure, reducing inflammation, attenuating cell death, and increasing bacterial killing. Cytomix-pre-activated MSCs are superior to naïve and hypoxia-exposed MSCs in attenuating *Klebsiella* pneumosepsis, improving lung compliance and oxygenation, reducing bacteria, and attenuating histologic injuries in lungs. BAL inflammatory cytokines were reduced, correlating with decreases in polymorphonuclear (PMN) cells. MSCs increased PMN apoptosis and the CD4:CD8 ratio in BAL. Systemically, granulocytes, classical monocytes, and the CD4:CD8 ratio were reduced, and nonclassical monocytes were increased. **Conclusions**: Preconditioning with cytokines, but not hypoxia, enhances the therapeutic potential of MSCs in clinically relevant models of *K. pneumoniae*-induced pneumosepsis.

## 1. Introduction

The emergence of antimicrobial-resistant (AMR) and hypervirulent bacterial strains is becoming an ever more prevalent problem in the treatment of infections, posing a very high risk to human health [1]. The incidence of hospital-acquired infections has been reported to reach 15% of patients with up to a 30% mortality rate in those that contract AMR strains [2], and it is estimated that, if left unaddressed, 10 million deaths per year will be attributable to AMR infections by the year 2050 [3].

Mesenchymal stromal cells (MSCs) have previously been shown to have potent anti-bacterial properties [4,5] along with antiviral [6] and antiparasitic [7] effects, both directly affecting microbial infections and indirectly through their immunomodulatory properties. Directly, the MSCs release antimicrobial peptides, and this can be enhanced through preconditioning the cells with bacterial compounds and inflammatory cytokines [8].

MSCs have been demonstrated to be an effective therapeutic strategy for the treatment of pneumonia [4] and ventilator-induced lung injuries (VILI) [9] in animal models due to their anti-inflammatory and immunomodulatory effects. MSCs’ mechanism of action has been demonstrated in these models to be mediated by means of their ability to reduce the initial acute inflammatory response to injuries and infections, reducing further tissue damage and elevating immune responses that reduce the bacterial burden. However, patients are rarely admitted to the hospital at this early stage of infection. It is evident that better later-phase models of more established infections are needed for testing the efficacy of MSC therapeutics in sepsis-induced ARDS to better replicate the clinical condition.

Despite the safety and trend toward efficacy demonstrated by MSCs in ARDS clinical trials, such as the START trials [10], it is apparent that the therapy needs to be made more efficacious. It is more feasible to enhance the MSCs’ function rather than further increase the doses of MSCs administered, which could incur the risk of infusional toxicities and even frank pulmonary embolism and also increase the costs involved with the generation of large cell doses [11]. This enhancement can be achieved by either genetic or nongenetic modification strategies [12]. Choosing nongenetic modification avoids the risk of virally elicited immune responses and genome integration, which could possibly lead to tumorigenesis [13]. Nongenetic modification methods involve the pre-activation or preconditioning of the MSC culture conditions to induce a phenotypical change [14]. This can be performed by physical, chemical, or environmental stimulation with the principle of creating a ‘primed’ MSC, which would be more reactive to the in vivo environment to which they are directed. 

Established pneumonia is characterised by high levels of ongoing inflammation and a significant impairment to the alveolar epithelium and endothelium, reducing oxygen exchange [15,16]. It has been shown that MSCs home to sites of injury and inflammation [17], and our previous studies have demonstrated that, regardless of the inflammation state, IV-administered MSCs become lodged in the lung vasculature immediately after administration [18]. The MSCs are cleared within 24 h in healthy lungs but remain there for over 24 h in states of active infection. Therefore, the administration of MSCs to treat pneumonia would mean that they enter into a climate with a high inflammatory cytokine concentration and low oxygen levels. Consequently, in this study, we elected to precondition MSCs by exposing them to a cytokine cocktail or hypoxic conditions. 

Our first hypothesis is that the application of MSCs to established pneumonia induced by an AMR *Klebsiella* pathogen would improve physiological and immunological parameters in vitro and in vivo. Our second hypothesis was that the application of either or both cytokine and hypoxic preconditioning would further enhance the MSC therapeutic potential in AMR pathogen-induced established pneumonia.

## 2. Results

### 2.1. Pre-Activation Enhances the Functionality of UC-MSCs In Vitro

Pre-activated MSCs have a superior effect over naïve MSCs in in vitro models of wound healing and inflammation. By simulating the different injury and repair processes seen in the lung over the course of a pneumonia injury, it was hoped to model the direct effects of our MSC therapy in vitro. Pulmonary epithelial wound healing was enhanced following treatment with the conditioned media (CM) from pre-activated and naïve MSCs, with cytomix pre-activation significantly enhancing wound closure compared to the other treatment types at 3 h, although this difference was not evident at later timepoints (Figure 1A). At 22 h, naïve MSCs had a significantly better effect than the vehicle, which was lost at subsequent timepoints. Cytomix-activated MSCs showed significant increases in wound healing at all timepoints following wound creation. Hypoxia pre-activation demonstrated this effect at 11 h and 22 h, with the effect lost at later timepoints compared to the vehicle (Figure 1A). 

Treatment with both naïve and pre-activated MSC-CM reduced the inflammation induced by cytokine injury (Figure 1B) and LPS injury (Appendix A) to a comparable degree, with hypoxia pre-activation only reducing TNF-α secretion on average (Appendix A). All MSC-CM improved cell viability following peroxide injury (Figure 1C). MSC-CM treatment derived from hypoxia-exposed MSCs showed superior effects over both naïve and cytomix MSCs (Figure 1C). Cytomix-pre-activated MSC-CM enhanced the neutrophil-like expression of CD11b, while hypoxia MSC-CM significantly reduced their expression (Appendix A). The pre-activation of MSCs enhanced their ability to reduce the number of viable bacteria in cultures of *E. coli*, *K. pneumoniae*, and *S. aureus* (Figure 1D–F). Hypoxia-treated MSC-CM had a greater effect than naïve and cytomix-activated MSCs against both gram-positive and gram-negative strains.

### 2.2. Pre-Activation of MSCs Attenautes Physiological Dysfunction in Established Klebsiella pneumosepsis

Sixty animals (twelve per group) were entered into this study. All animals survived the *K. pneumoniae* intratracheal instillation which induced a significant lung injury compared to the sham animals at the 72 h endpoint (Figure 2).

The intravenous (IV) administration of cytomix-pre-activated MSCs via the tail vein to rodent models of established pneumonia attenuated indices of lung injuries and infections. MSCs were characterised according to ISCT surface marker expressions (Appendix A), showing a reduced CD105 expression for hypoxia MSCs, while cytomix had an increased HLA-DR expression. Cytomix-pre-activated MSCs significantly diminished the decrease in lung compliance seen following *K. pneumoniae* infection (Figure 2A). Similarly, the increased serum lactate levels indicating impaired tissue oxygenation [19] and sepsis [20] was lowered significantly after the administration of cytomix-pretreated MSCs (Figure 2B). Treatment with cytomix-pre-activated MSCs significantly reduced bacterial colony forming units (CFUs) in the BAL fluid (Figure 2C). The amounts of infiltrating white cells were also lowered by both naïve and cytomix-pre-activated MSCs (Figure 2D). Interestingly, hypoxia-conditioned MSCs appeared to be less effective in most of these parameters, significantly so in terms of white cell infiltrates when compared to the naïve MSC group (Figure 2D).

### 2.3. Naïve and Cytomix-Exposed MSCs Attenuate K. pneumonia-Induced Histologic Injury

Cytomix-pre-activated MSC therapy—but not naïve or hypoxic-pre-activated MSCs—improved the airspace fraction evident in the lungs compared to the vehicle control. There were less infiltrates and a reduced alveolar wall thickening evident in groups that received cytomix-pretreated MSCs (Figure 3A–F). 

### 2.4. MSCs Differentually Attenuate the Pulmonary Cytokine Response to K. penumonia Infection

The levels of inflammatory cytokines IL-1β, CINC-1, and TNF-α were significantly reduced in animals that received both naïve and cytomix-pre-activated MSCs (Figure 4A–C). IL-18 and IL-12 were significantly increased in the BAL of cytomix and hypoxia MSC-treated animals (Figure 4D,E), while only the hypoxia MSC-treated group showed no reduction in IFN-γ levels in the BAL compared to the vehicle control (Figure 5C). MIP-1α and MIP-3α were reduced in the BAL of naïve and cytomix MSC-treated animals (Figure 4F,G). Both naïve and cytomix MSC-treated animals had a significant reduction in MMP-9 and MCP-1 in the BAL (Figure 4H,I). Taken together, there is an enhanced resolution of infection with naïve and cytomix-licensed MSC administration leading to a decrease in inflammatory cytokine secretion by the lungs.

IL-4 was at low levels in the BAL for all groups, but there was a significant increase in the cytomix and hypoxia MSC-treated animals (Figure 5A). Both the cytomix and hypoxia MSC-treated animals had significantly elevated levels of IL-2, while naïve MSCs significantly lowered the amount of IL-2 in the BAL (Figure 5B). IFN-γ, a proinflammatory cytokine produced by Th1 cells, was significantly reduced in the BAL of naïve and cytomix MSC-treated animals (Figure 5C). RANTES was lower with the naïve MSC treatment with no observable change with the other treatments when compared to the vehicle control (Figure 5D). There were elevated levels of anti-inflammatory IL-10 in the BAL, with naïve MSCs significantly lowering the amount present (Figure 5E). IL-6 levels in the BAL were reduced with naïve MSC administration and were significantly lower in cytomix-treated animals (Figure 5F).

### 2.5. BAL White Cell Subsets Indicate a Resolution of Pneumonia Following MSC Administration

Neutrophils play a crucial role in the clearance of invading pathogens. The administration of naïve and cytomix-activated MSCs significantly reduced neutrophil numbers in the BAL with both treatments being significantly better than hypoxia-exposed MSCs (Figure 6A). Anti-granulocyte RP-1^+^ neutrophils from naïve- and cytomix-treated animals also have a proapoptotic phenotype compared to the vehicle treatment, attributing to their reduced numbers in the BAL (Figure 6B). There was a significant decrease in the level of the CD11b/c activation marker expression on the neutrophils in the hypoxia MSC treatment group (Figure 6C). MSC administration did not significantly change the total number of CD4^+^ T helper cells or CD8^+^ cytotoxic T cells in the BAL (Appendix A). However, there was a significant increase in the CD4:CD8 ratio for all treatment groups (Figure 6D). The sham data was not included due to the sparsity of neutrophils in the BAL of healthy rodents.

### 2.6. BAL Macrophages/Monocytes Show Enhanced Cell Function Following MSC Administration

Macrophages are the first responders of the immune system and play an important role in maintaining local and systemic immune homeostasis. After *K. pneumoniae* injury, there was no change in the total number of macrophages/monocytes in the BAL between treatment groups (Figure 7A), with these cells showing indices of immunoparalysis compared to the sham (Figure 7B,C). The naïve and cytomix-pre-activated MSC treatment significantly enhanced BAL macrophage/monocyte phagocytosis and superoxide anion production, with no observable change occurring after hypoxia MSC administration (Figure 7B,C). After an ex vivo *K. pneumoniae* LPS injury for 18 h, there was over a 40% reduction in the vehicle macrophage/monocyte cell function, while naïve and cytomix-licensed MSC therapy significantly enhanced both phagocytosis and superoxide anion production (Appendix A). Hypoxia-grown MSCs only showed improved phagocytosis after the LPS injury (Appendix A). Assessing the secretome from these cells post-LPS exposure showed that naïve MSC administration improved the secretion of TNFα, IL-1β, IL-10, and IL-6, with IL-10 reaching significance (Appendix A).

### 2.7. Systemic Leukocyte and Lymphocyte Number and Function Are Altered by the Administration of Naïve and Cytomix-Pre-Activated MSCs

The levels of circulating monocytes were increased in animals which received MSC treatment (Figure 8A), and there were no changes in the circulating lymphocyte number (Appendix A). Naïve and cytomix-pre-activated MSCs significantly reduced the proportion of classical CD43^-^ monocytes and increased the level of CD43^+^ monocytes in the blood (Figure 8B,C). The CD4 to CD8 ratio decreased in cytomix- and hypoxia-exposed MSC-treated animals (Figure 8D), while the levels of T regulatory cells increased but did not reach significance in any MSC-treated groups (Appendix A). The sham monocyte populations were not assessed due to limited sampling from healthy controls.

The administration of cytomix-pre-activated MSCs resulted in a significantly reduced number of circulating neutrophils, while a significantly increased number was observed in animals treated with hypoxia-pre-activated MSCs (Figure 9A). All MSC types resulted in slightly increased neutrophil apoptosis and significantly reduced neutrophil activation (Figure 9B,C). The sham CD11B/C median fluorescent intensity (MFI) was not included as we could not perform a relative fold change for that sample.

## 3. Discussion

Pneumonia is a serious respiratory illness and has a high mortality rate in patients admitted to the ICU [21]. Antimicrobial resistance remains a prevalent problem in the treatment and management of bacterial pneumonia [22]. MSCs have been observed to be a potential treatment for bacterial pneumonia due to their immune modulating capacity, and this has been clearly demonstrated in preclinical models in recent years [23,24,25]. Despite this promise shown in laboratory testing, the efficacy has not been fully translated to human trials [26]. This could be due to several factors, including requirements for a more potent cell therapy. We demonstrated in this study that the effects of naïve MSCs can be enhanced using a variety of different pre-activation strategies. When translating the most promising MSC pre-activation strategy to preclinical models, we first need to demonstrate an enhanced effect in vitro. However, our in vitro assessment of these pre-activation strategies demonstrated comparable efficacy in all assays used. Characterisation of the cells post-cryopreservation showed a large population of CD105 MSCs after hypoxia pre-activation. This population of cells has been associated with enhanced immunosuppression [27] as shown by the reduced CD11b activation on neutrophil-like cells, yet they were less effective in modulating the macrophage-like inflammatory response. Therefore, the initial finding of this study is that the translation from in vitro to in vivo is not clear cut. We have previously demonstrated that in vitro assays are able to predict our cell therapy efficacy in rodent models [28], which is why wound-closure, NF-kappa-B activation, and ROS injury assays were included. However, refinement of these assays is needed and is currently being addressed in other models and systems [29]. 

The promise of cytomix pre-activation as a potent MSC enhancement technique has been previously demonstrated in vitro [30] and in other ARDS models [31]. The physiological indices of injury were improved with the application of cytomix-pre-activated MSCs in our model of pneumonia including an increase in static lung compliance, a decrease in serum lactate, a decrease in live bacteria in the lung, and a reduction in inflammatory white cells to the lung airspace. Naïve MSCs had no significant effect in many of the physiological parameters, unlike what we have shown previously in our other models of infection and injury. As here we have used a clinically isolated β-lactamase-producing multi-drug resistant strain of *Klebsiella pneumoniae* with a hypermucoviscous phenotype, this data implies that the strain of bacteria involved in the development and progression of pneumonia will need to be considered in the treatment strategy and shows the importance of extensive research into pneumonia of different etiologies. 

Inflammatory cytokines in the BAL fluid were decreased with naïve and cytomix MSC treatment as has been demonstrated previously, but notably, these were not changed in most cases after hypoxia-pretreated MSC administration compared to no treatment. This correlates with the presence of PMNs and neutrophils and the CD4:CD8 T cell ratio which indicates that MSC treatment confers an innate and adaptive immune response associated with recovery and repair [32,33]. These cytokines are indicative of the inflammatory state and correlate with the severity of pneumonia, mortality [34,35], and with the bacterial burden [36]. IL-18 along with IL-12 direct cell-mediated immunity as IL-12 induces the differentiation of naïve CD4^+^ T cells to T helpers of type 1, while IL-18, a proinflammatory cytokine of type I and II inflammation, can induce the production of IFN-γ from these cells and promotes IL-1 pathways in critically ill patients. IFN-γ primes macrophages for an inflammatory response, and should LPS be present, it promotes tumoricidal, bactericidal, and inflammatory cytokine production thereby enhancing their function [37,38]. 

Both MIP-1α and MIP-3α are powerful chemoattractants, with MIP-1α primarily attracting monocytes/macrophages but also preferentially attracting CD8 T cells, while MIP-3α attracts memory T cells, naïve B-cells, immature dendritic cells, and, at high enough concentrations, neutrophils [39]. In our animal model, we can see a reduction in MIP-1α and a significant reduction in MIP-3α in the BAL of naïve- and cytomix-treated animals partially accounting for the decreased leukocyte infiltration into the alveolar airspace. In the BAL of both naïve- and cytomix-treated animals there was a significant reduction in the level of MMP-9. MMP-9 is involved in the degradation of the basement membrane extracellular matrix and is released by neutrophils in the presence of inflammatory mediators such as IL-8/CINC1 and TNF-α. It is functionally required for neutrophil migration into the alveolar airspace and causes capillary leakage leading to oedema formation [40]. MCP-1 is a potent chemokine associated with monocyte recruitment and is predominantly secreted by macrophages. It has been identified as a component of increased oxidative stress and interplays with other cytokines by suppressing IL-12 secretion in dendritic cells thereby inhibiting naïve CD4^+^ T cell differentiation into Th1 cells [41]. In our animal model, MCP-1 was significantly reduced in the BAL of both naïve- and cytomix-treated animals. Taken together, there is an enhanced resolution of infection with naïve and cytomix-licensed MSC administration, leading to a decrease in inflammatory cytokine secretion by the lungs.

IL-4 has been studied extensively and promotes the differentiation of naïve CD4^+^ T cells into T helper cells of type 2, promotes the proliferation of antigen-specific CD8^+^ T cells, and shows anti-inflammatory effects on LPS-injured monocytes by suppressing TNF-α secretion [42,43]. IL-4 was at low levels in the BAL for all groups, but there was a significant increase in cytomix and hypoxia MSC-treated animals. IL-2 is regulated by antigen recognition and costimulatory molecules. It is produced primarily by effector memory T cells in vivo and promotes survival and the proliferation of T cells [44]. Both the cytomix and hypoxia MSC-treated animals had significantly elevated levels of IL-2 in the BAL, while naïve significantly lowered the amount of IL-2 in the BAL. As mentioned previously, IFN-γ is a proinflammatory cytokine produced by Th1 cells, and we can see a significant reduction in IFN-γ levels in the BAL of naïve- and cytomix-treated animals. RANTES is primarily associated with type II inflammation by promoting leukocyte migration, primarily eosinophils, lymphocytes, neutrophils, and monocytes. However, this is controversial as studies have shown Th2 to Th1 switching with RANTES and increased IFN-γ secretion [45]. RANTES in the lungs of our infected animals was lower with naïve MSC treatment, with no observable change in the other treatments when compared to the vehicle control. IL-10 is a potent anti-inflammatory cytokine known to be produced by nearly all activated immune cells, from granulocytes to lymphocytes. It does this by reducing proinflammatory cytokine secretion, limiting antigen presentation, and inhibiting macrophage activation and infiltration [46]. There were elevated levels of IL-10 in the BAL. Naïve MSCs significantly lowered the amount of IL-10 present which may be due to the resolution of the infection already occurring. IL-6 is an intermediatory between the innate and adaptive immune system, showing both pro- and anti-inflammatory properties. IL-6 is a pleiotropic cytokine, with roles including iron sequestering, increased platelet production, differentiation of naïve CD4^+^ T cells, and increased vascular permeability, and is crucial in the switch from neutrophil infiltration to monocyte and T cell infiltration during infections [47,48]. IL-6 levels in the BAL were reduced with naïve MSC administration and were significantly lower in cytomix MSC-treated animals.

Increased ratios of CD4 to CD8 T cells in the BAL are associated with sarcoidosis. While these levels typically exceed 3.5 in the BAL and so cannot be approximated to this animal model, it should be taken into consideration should sarcoidosis be a comorbidity in sepsis patients receiving MSC therapy. Current studies in MSC therapy for sarcoidosis focus on the phenotype shift from M1 to M2 alveolar macrophage functions [49], and so, further investigations are needed especially considering the findings portrayed in this study. Another study on infant acute lung injury during viral infection showed that there is a correlation between a reduced CD4^+^ to CD8^+^ ratio and lung damage and so brings credence to the clinical trials involving MSCs as a therapeutic for COVID-19 infections [50]. Low systemic CD4 to CD8 T cell ratios are indices of immune senescence, chronic inflammation, and altered cellular functions, all of which increase tissue damage and indicate a poor prognosis [51]. This would partially explain the significantly improved function of BAL macrophages in the naïve and cytomix MSC treatment groups but not the hypoxia group, which did trend towards an improved functionality but did not reach significance, and so, other factors are at play.

Monocytes are antigen-presenting cells (APCs) and are part of the professional phagocytes within the innate immune system. They play an important role in maintaining tissue and immune homeostasis [52]. Classical monocytes have an enhanced phagocytic capacity and are primed for migration and immune cell signalling [53]. Nonclassical monocytes patrol the endothelium in search of injury, antigen presentation to T cells, and partake in complement and Fc gamma-mediated phagocytosis and adhesion [53]. Rat monocyte subpopulations can be identified based on their level of CD43 expression [54]. Monocytes have an increased survival, phagocytosis, activation, APC potential, and adapt a regulatory phenotype in the presence of MSCs [55,56,57]. Previous literature describes MSCs’ ability to reprogram inflammatory classical to anti-inflammatory nonclassical monocytes leading to overall therapeutical benefits in animal models in part mediated by the suppression of T cell proliferation [58]. Here we demonstrated that both naïve and cytomix-pre-activated MSC treatments significantly increased the proportion of nonclassical monocytes to classical ones in circulation, which may partially explain the improved lung function and structure after cytomix pre-activation. However, further research is needed into their activation state and function after MSC treatment as this can drastically alter their role [59].

Neutrophils are a crucial cell type needed for the rapid clearance of invading pathogens; however, their dysregulation is prevalent in hyperinflammatory diseases such as pneumonia. This dysregulation results in delayed apoptosis and increased numbers of systemic and local neutrophils coupled with persistent overactivation. These neutrophils are responsible for prolonged tissue damage due to the release of granzyme proteins and reactive oxygen species (ROS) [60]. For sepsis patients, the antiapoptotic phenotype denoted by neutrophils is detrimental to their survival chances and morbidity [61]. In this study, we have shown a reduced systemic neutrophil number and an adhesion/activation marker expression which partially accounts for the reduced neutrophil infiltration into the BAL. Since naïve and cytomix-licensed MSCs reduce inflammation in the alveolar airspace, this contributes to the restoration of a proapoptotic phenotype compared to the vehicle control. Apoptotic neutrophils also play an important role in inducing anti-inflammatory alveolar macrophages to an M2 phenotype, which is crucial for disease resolution [62], and we see an increased M2 phenotype with naïve administration with an increased IL-10 secretion in the presence of an endotoxin injury. 

A theme permeates throughout the series, with hypoxia MSC treatment either not being different from the vehicle treatment or divergent to the other treatment groups. This divergence occurs in BAL neutrophil activation, similar to what was seen in an inflammatory environment in vitro. While hypoxia MSC secretome was most effective in reducing the proliferative capacity of gram-negative and gram-positive pathogens, thought to be due to antimicrobial peptides [8], both in vitro and in vivo analyses showed a suppressed neutrophil activation state. Along with the impaired bacterial killing capabilities of adherent BAL macrophages/monocytes, this may have allowed for an initial uncontrolled growth of the K. pneumoniae accounting for the reduced lung function. Studies have also shown an improved survival of hypoxia-grown MSCs after transplantation which may have further exasperated this issue due to increased immunosuppression [63] along with impaired transition from M1 to M2 macrophage phenotypes, preventing the resolution to infection [64]. These hypoxic MSCs may be better suited for autoimmune disorders, but further research is needed.

All things considered, this bodes well for MSC therapies as it indicates a reduced influx of neutrophils from the bone marrow that leads to systemic organ damage and neutropenia while also reducing the antiapoptotic phenotype that causes long-term morbidity in sepsis patients. In recovering patients, there is a risk that, due to an exhausted or over-exposed immune system, they will succumb to a secondary infection [65,66]. Here we demonstrated that macrophages isolated from the lung and exposed to a secondary injury can respond accordingly after the animals they came from received MSC treatment. This was especially true of their phagocytic index and ability to produce superoxide anions and did not occur with hypoxia conditioning. The cytokine secretion of these cells in response to an endotoxin injury was unchanged for inflammatory cytokines, while anti-inflammatory IL-10 was significantly increased with naïve MSC treatment.

In summary, we have demonstrated here that, firstly, a more established and therefore more difficult to treat pneumosepsis does not respond to conventional MSC therapies as previously reported for short-term, acute infection models. Secondly, the pre-activation of MSCs using cytomix seems to be necessary for the therapeutic benefits to be realized in this model. Thirdly, the commonly used preconditioning method of hypoxia exposure is, in fact, a poor method of MSC enhancement for use in this model and should be brought forward with caution. Future studies should build on these results and address other factors such as the route of administration, the optimal MSC or MSC product, and different methods of cell preservation to enable a better understanding of the mechanisms of action of MSC therapy.

## 4. Materials and Methods

### 4.1. Human Umbilical Cord-Derived MSCs

Human umbilical cord (hUC) perivascular MSC cell populations were provided by Tissue Regeneration Therapeutics (Toronto, ON, Canada). The UC-hMSCs were thawed and expanded as previously described [28]. Naïve and activated hMSC populations were cultured at 37 °C, 95% humidity, and 5% CO_2_ until 70–80% confluent and then were typsinised and culture-expanded to passage 3–4. MSCs were exposed to the following pre-activators: the vehicle (naïve) and cytomix (IL-1β [50 ng/mL], TNF-α [50 ng/mL], and IFN-ɣ [50 ng/mL], all from Immunotools Ltd., Friesoythe, Germany), for 24 h and then were either typsinised, resuspended and cryopreserved in 15% DMSO and 90% FBS (both Sigma-Aldrich, Dublin, Ireland), or used to generate conditioned media (CM) as previously described [31]. Hypoxic-grown hMSC populations were cultured at 37 °C, 95% humidity, 5% CO_2_, and 2% O_2_ from passage 1 until the end of passage 4 and then were either typsinised, resuspended and cryopreserved, or used to generate conditioned media (CM). Immediately before administration to the animal models, cryovials containing 10^7^ cells were thawed, and trypan blue exclusion dye viability staining was performed. Ten million MSCs per kg of animal weight were pelleted at 400× *g* for 5 minutes and were resuspended in 1mL of sterile PBS and were placed on ice until administration. 

### 4.2. In Vitro Assay Panels to Ascertain MSC Functionality

#### 4.2.1. Wound-Healing Assay

Alveolar epithelial cells (A549 cell line, ATCC) were grown to confluence in 24-well plates (Thermo Scientific™ Nunc™, Dublin, Ireland). A single vertical wound was created in the monolayer using a P1000 tip, and the media was changed to remove cells in suspension. Cell layers were exposed to CM from either naïve or pre-activated MSCs (cytomix or hypoxia) at a ratio of 3:1 with standard cell line media and were incubated under normal culture conditions. Wound sizes were measured at timepoints between 0 h and 40 h post-creation using a live cell imager (Cytation, Biotek Ltd., Bad Friedrichshall, Germany).

#### 4.2.2. Nuclear Factor κB Activation Assay

A549 cells stably transfected with a κB-luciferase reporter construct (Thermo Fisher, Waltham, MA, USA) were grown to confluence in 96-well plates (Thermo Scientific™ Nunc™). Cell monolayers were subjected to inflammatory cytomix injuries (IL-1β [50 ng/mL], TNF-α [50 ng/mL], and IFN-ɣ [50 ng/mL], all from Immunotools Ltd.) and then were treated with the vehicle or conditioned medium (CM) from the vehicle, cytomix-, or hypoxia-exposed UC-MSCs or the vehicle. Cells were harvested at 24 h and assayed for luciferase content as an indicator of NF-κB activation.

#### 4.2.3. Cell Metabolic Activity

The MTT assay was performed using the MTT reagent ((3-(4, 5-dimethylthiazol-2-yl)-2,5-diphenyltetrazolium bromide, Sigma Aldrich Ltd., Wicklow, Ireland) to evaluate cell metabolic activity as an index of viability. Bronchial epithelial (BEAS2B) cell monolayers were injured using 4 mM hydrogen peroxide and then were treated with UC-MSC-CM isolated from each pre-activated MSC or vehicle. A total of 4 h after treatment, BEAS2B cells were washed with PBS followed by incubation with the MTT reagent for 4 h at 37 °C in a humidified cell culture incubator. The resultant formazan crystals were solubilized using dimethyl sulfoxide, and absorbance readings were measured using the Varioskan™ Flash microplate reader (Thermo Fisher Ltd., Waltham, MA, USA) at a 595 nm wavelength. The degree of cell metabolic activity was presented as a percentage relative to the vehicle control.

#### 4.2.4. Inflammatory Cytokine Production

Inflammatory, anti-inflammatory, and other functional cytokine and growth factor levels in blood plasma and BAL fluid were measured using a 23-cytokine multiplex assay (Bio-Rad, Naas, Ireland) or standard ELISAs (R&D systems, Minneapolis, MN, USA).

#### 4.2.5. Bacterial-Killing Assay

Viable bacterial cultures of *E. coli*, *K. pneumonia*, and *S. aureus* were grown in the MSC-CM generated from naïve, cytomix-, and hypoxia-pretreated MSCs for 6 h following the protocol previously described [67]. Serial plate counts and optical density readings (OD 590 nm) were performed to ascertain if the bacterial-killing capacity of the MSCs was enhanced by the pre-activation methods. 

### 4.3. Preclinical Experimental Series

All work was approved by the Animal Care in Research Ethics Committee of the National University of Ireland, Galway and was conducted under a license from the Health Products Regulatory Agency, Ireland (AE19125/P067). Specific pathogen-free adult male Sprague Dawley rats (Envigo, UK) weighing between 350 g and 450 g were used in all experiments. Animal welfare was monitored at regular intervals and distress scoring was used to determine if the predicted severity exceeded at any point. All animals entered into the experiments were administered regular buprenorphine analgesia (0.03 mg.kg^−1^, Bupaq, Chanelle, Galway, Ireland).

#### 4.3.1. Established Pneumonia

Adult male Sprague Dawley rats were anesthetized by isoflurane inhalation. In total, 5 × 10^8^ viable CFUs of *Klebsiella pneumoniae* were administered in a 300 µL bolus intratracheally under direct vision. Animals were allowed to recover from the anesthesia and were rehoused. The animals were monitored and their statuses, including behavioral signs and welfare, were recorded for 72 h after pneumonia induction.

#### 4.3.2. Experimental Design

Animals were randomized to receive the intravenous (IV) PBS vehicle or 10^7^ UC-MSC.kg^−1^, naïve or pre-activated, with either cytomix or hypoxia 1 h after pneumonia induction (*n* = 12 per group). The degree of injury was assessed at 72 h.

#### 4.3.3. Premortem Assessment

At 72 h post-pneumonia induction, the animals were anesthetized with subcutaneous ketamine (75 mg.kg^−1^, Ketamidor™; Chanelle, Galway, Ireland) and medetomidine (0.5 mg.kg^−1^, Medetor™, Chanelle, Galway, Ireland). When the rodents were unconscious, the depth of anesthesia determined by a paw clamp, an IV cannula was sited and secured in the tail vein. A surgical tracheostomy was performed, and a 12 G tracheostomy tube was secured. Intra-arterial access for blood sample analysis and monitoring was gained using a 23 G cannula in the right carotid artery. Anesthesia was maintained with alfaxalone (2 mg.kg^−1^, Alfaxan™; Vetoquinol Ltd., Towcester, UK) and mechanical ventilation was commenced at a 7 mL.kg^−1^ tidal volume. Arterial blood gas analysis was performed (ABL Flex 90, Radiometer, O’Callaghan’s Mills, Co. Clare, Ireland), and blood cell counts were obtained using a hemoanalyser (Mythic 18, Orphee, Switzerland).

### 4.4. Postmortem Assessment

After exsanguination under anesthesia, blood and bronchoalveolar lavage (BAL) samples were collected for cytokine profiles and bacterial load measurements. Whole blood was divided and either centrifuged to yield plasma samples or subjected to the density gradient separation of peripheral blood mononuclear cells (PBMCs) and granulocytes using the histopaque^TM^ 1077 and 1119, respectively (both Sigma-Aldrich, Dublin, Ireland). 

#### 4.4.1. Bacterial Load

BAL fluid and whole blood were plated onto differential brilliance clarity UTI agar plates (Fannin Ltd., Galway, Ireland) and were incubated overnight at 37 °C. The total colony numbers of each indicative colour were counted. 

#### 4.4.2. Inflammatory Cytokine Profile

The cytokine-induced neutrophil chemoattractant (CINC-1), the tumor necrosis factor (TNF) α, and interleukin 6 (IL-6) were quantified by ELISA (R&D Systems), and 23 other cytokines and growth factors were measured using a multiplex immunoassay system (Bio-Plex Pro Rat Cytokine, Chemokine and Growth Factor Assay; Bio-Rad Ltd., Watford, UK).

#### 4.4.3. Histological Analyses

The left lung lobe was inflated using a 4% paraformaldehyde (PFA) solution and was tied off. The lobe was suspended in 4% PFA until use for histological analysis whereby the lobe was divided into 5 pieces craniocaudally, processed, and paraffin-embedded. Each lobe piece was sectioned into 7 μm thick cross-sections and stained using hematoxylin and eosin (H&E). Stereology was performed on images of each lung section taken at 20× and overlaid with a 10 × 10 grid. The intersections at each 50 μm space were scored as tissue, airspace, or nonacinar tissue to determine the percentage of airspace in the lung.

#### 4.4.4. Flow Cytometry Analysis of PBMCs

Freshly prepared PBMCs were resuspended in FACS buffer (PBS, 2% fetal calf serum & 0.05% sodium azide, Merck, Ireland) and were labelled with the following combinations of fluorochrome-labelled antibodies at 4 °C for 30 min: monocytes panel–anti-CD11BC PerCP/Cyanine5.5 (Clone:OX-42), CD43 PE (Clone: REA503), CD45 Vioblue (clone:REA504), and CD3 Viogreen (clone:REA223) (all from Miltenyi Biotec, North Rhine-Westphalia, Germany). Regulatory T cells were assessed via the intracellular labelling of FOXP3 according to the manufacturer’s instructions (Invitrogen, Waltham, MA, United States). Briefly, PBMCs were initially labelled with the cell surface markers anti-CD3 Viogreen (clone: REA223), CD4 PCP-vio700 (clone:REA489), and CD8b Vioblue (clone:REA222) from Miltenyi Biotec, Ireland and CD25 AlexFlour (clone:OX-39, BIO-RAD, Ireland) antibodies at 4 °C for 30 min. Cells were subsequently permeabilized and labelled with anti-FOXP3 PE (clone:150D, Biolegend, Ireland) and were fixed prior to analysis. Cell viability was analyzed using DRAQ-7 and fixable Ghost dye Red780 dead cell stain (Biolegend, San Diego, CA, USA) according to the manufacturer’s instructions. For all flow cytometry analyses, labelled cells were washed and resuspended in FACS buffer and immediately analyzed on an FACS Canto II cytometer (BD Biosciences). Data files were subsequently analyzed using FlowJo v10 software (Ashland, OR, USA). Details and examples of the gating strategies used to define and enumerate specific immune cell subpopulations are provided in Appendix A.

#### 4.4.5. Flow Cytometry Analysis of Neutrophils

Neutrophils were analyzed from 3% dextran sedimented whole blood and BAL samples. In brief, whole blood was mixed 1:1 with 3% dextran (Sigma) *w/v* in DPBS, and the solution was allowed to stand for 20 min before the upper buffy coat layer was isolated and pelleted. 

To remove contaminating RBCs, the WBC pellet and BAL cell pellet were resuspended in a 1× ammonium–chloride–potassium (ACK) lysis solution (10x ACK lysis buffer, 500 mL of distilled water, 40.1 g of NH4Cl, 5 g of KHCO3, and 190 mg of Na_2_-EDTA were brought to 1× in sterile water) for exactly 5 min before flooding the tube with complete rat media to halt the lysis reaction and wash the cells twice in complete rat media. BAL and whole blood WBCs were incubated for 1 h at 37 °C in an orbital shaker set to 180 rpm to allow the cells to become metabolically active. Whole blood WBCs and BAL cells were resuspended in 10% FACS buffer and were labelled with the following combinations of fluorochrome-labelled antibodies at 4 °C for 30 min: anti-neutrophils PE (clone RP-1), CD11b/c PE-vio770 (clone REA325), and Live/Dead™ Far Red (Invitrogen, Waltham, MA, USA). Cells were then washed and resuspended in a 1x Annexin-V binding buffer (10× solution consisting of 400 mL of distilled water, 9.53 g of HEPES, 32.73 g of NaCl_2_, and 1.11 g of CaCl_2_ with a 1:10 dilution in distilled water). A total of 5 µL of Annexin-V FITC (Biolegend) was added to each sample 10 min before analysis on the Accuri C6 flow cytometer.

#### 4.4.6. BAL Macro/Monocyte Phagocytosis and Superoxide Anion Production

A total of 2 × 10^5^ BAL cells were plated in a tissue culture-treated 96-well plate and were incubated for a minimum of 1 h to ensure adherence occurred and to start their metabolic activity. The adherent BAL cells were then either exposed to the PBS vehicle or *K. pneumoniae* LPS at 100 ng/mL for 18 h at 37 °C to assess their functionality before and after an ex vivo secondary injury. After this incubation period, the complete rat cell media was stored for later ELISA-mediated cytokine analysis. Zymosan A *S. cerevisiae* BioParticles™ (Invitrogen^TM^) opsonized with 2 mg/mL of human serum (Sigma) for 1 h at 37 °C were added at 8 particles/cell along with 0.3 µg/mL of DAPI and 0.35 mg/mL of NBT solution before being incubated for 30 min. Cells were then fixed with 4% PFA for 10 min before being stored in the dark at 4 °C. Analysis occurred on the Cytation 1, with images taken in brightfield, DAPI, and FITC before being overlayed for counting.

#### 4.4.7. BAL Macro/Monocyte Quantification

Differential BAL white cell counts were completed using a variant of the Romanowsky–Giemsa stain, Diff-Quik. Superfrost™ Plus microscope slides were mounted to a Shandon cytospin system and were placed into a Cytospin 4 chamber (all Thermo Fisher) before 150 µL of BALF was added to the funnel. Samples were spun at 200 RPM for 5 min before the slide was removed from the chamber. After air drying for 5 min, the slides were fixed and stained by immersing them 6 times in methanol, 5 times in Eosin Y, 3 times in methylene blue, 5 times in a buffered solution, and 5 times in a rinse solution before being left to dry. After drying, one drop of DPX mountant (VWR International, Dublin, Ireland) was placed on each section and a cover slip (Thermo Fisher) was placed over and allowed to dry. Stained cells were visualised under light microscopy using a BX43 Olympus microscope (Olympus Life Sciences, Tokyo, Japan), and images were taken with a GXCAM-EYE-5 microscope attachment (GT Vision Ltd. Sudbury, Newmarket, UK). Differential morphological analysis was performed on the images to determine monocytes/macrophages. A total of 300 cells across at least 3 images were analysed per sample.

### 4.5. Statistical Analyses

GraphPad Prism software (GraphPad Software Ltd., San Diego, CA, USA) was used to graph and statistically analyze the data. Kolmogorov–Smirnov tests were employed to determine if variables were normally distributed followed by analyses by one-way ANOVA and Tukey post hoc testing to correct for multiple comparisons. A two-tailed *p* value of less than 0.05 was considered significant.

## Figures and Tables

**Figure 1 pharmaceuticals-16-00149-f001:**
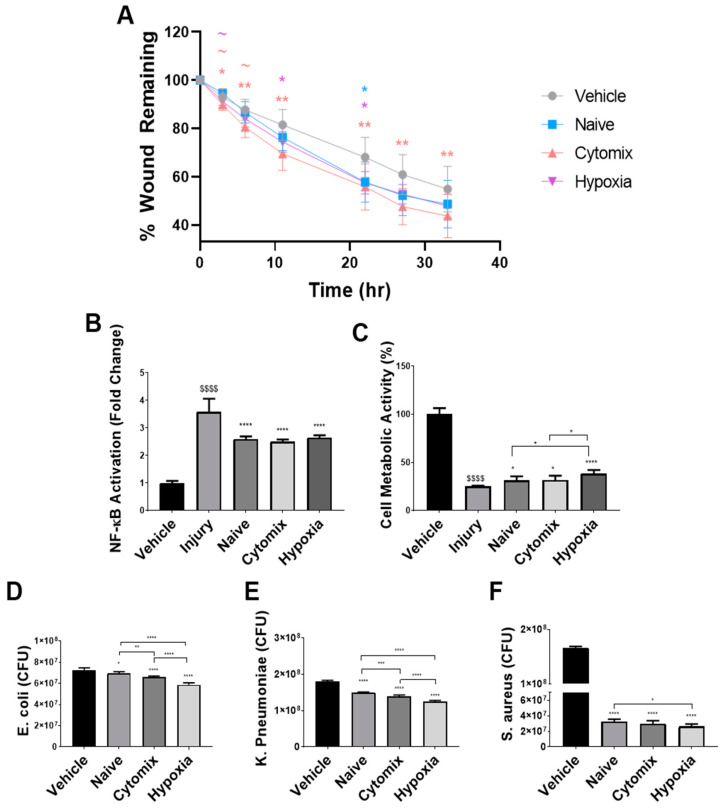
Pre-activated MSC-derived CM significantly increased the rate of wound healing (**A**) and attenuated the increase in inflammatory injuries induced by cytomix in A549s (**B**). Pre-activated MSC-CM also restored the decreased cell metabolic activity induced by peroxide injury (**C**) compared to the vehicle control. The application of conditioned media from naïve and pre-activated MSCs to cultures of bacteria significantly decreased the viable CFU of *E. coli* (**D**), *K. pneumoniae* (**E**), and *S. aureus* (**F**). CFU = colony forming units. All datapoint symbols represent individual datapoints, and columns represent means. Error bars represent SDs. *, **, ***, **** = *p* ≤ 0.05, 0.01, 0.001, 0.0001 with respect to the vehicle control. $$$$ = *p* ≤ 0.0001 with respect to the sham group. *N* = 6–12.

**Figure 2 pharmaceuticals-16-00149-f002:**
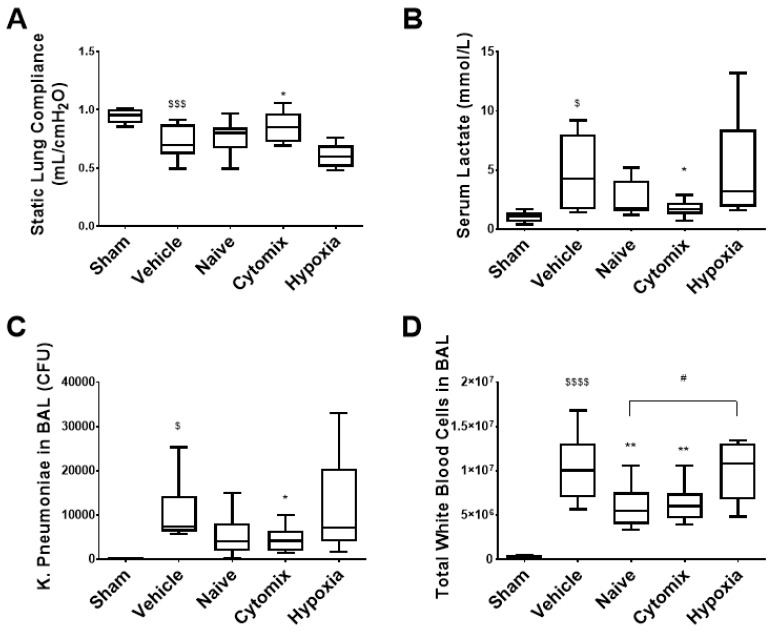
Respiratory static compliance was significantly improved with the application of cytomix-pre-activated MSCs and appeared to be worsened by the administration of hypoxia-exposed MSCs (**A**). The serum lactate was significantly reduced after the administration of cytomix-pre-activated MSCs compared to the vehicle control (**B**). Viable bacteria in the BAL fluid were significantly reduced in animals treated with cytomix-pre-activated MSCs (**C**). The number of pulmonary white blood cells was significantly elevated in the vehicle control group compared to the sham (**D**), whereas the naïve and cytomix-pre-activated MSC-treated groups had significantly lower white cell counts in the BAL than in the vehicle control. Hypoxia pretreatment did not improve the total white cell infiltration and was significantly worse than the naïve MSC treatment (**D**). CFU = colony forming units. Box plots and whiskers represent the minimum, first quartile, median, third quartile, and maximum. *, **, = *p* ≤ 0.05, 0.01 with respect to the vehicle control. $, $$$, $$$$ = *p* ≤ 0.05, 0.001, 0.0001 with respect to the sham group. # = *p* ≤ 0.05 with respect to other treatment. *N* = 8–12.

**Figure 3 pharmaceuticals-16-00149-f003:**
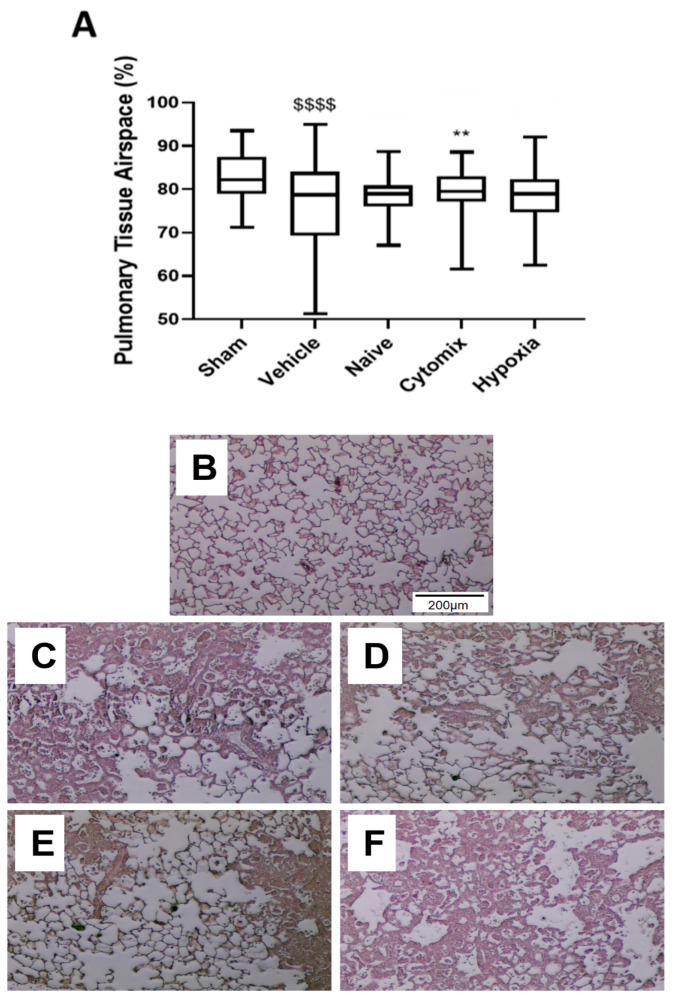
Sections of PFA-fixed, H&E-stained, and paraffin-embedded lung tissue were analyzed using stereology for the percentage of airspace present in the lungs (**A**–**F**). Representative images of the sham (**B**), vehicle (**C**), naïve (**D**), cytomix (**E**), and hypoxia (**F**) histology. Animals receiving cytomix-pretreated MSCs had a significantly higher airspace fraction compared to the vehicle control (**A**). Box plots and whiskers represent the minimum, first quartile, median, third quartile, and maximum. ** = *p* ≤ 0.01 with respect to the vehicle control. $$$$ = *p* ≤ 0.0001 with respect to the sham group. *N* = 8–12.

**Figure 4 pharmaceuticals-16-00149-f004:**
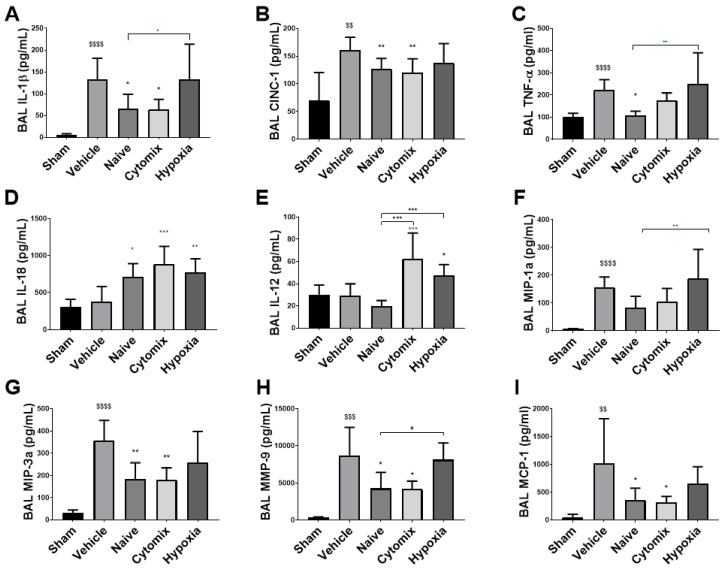
The inflammatory cytokines IL-1β, CINC-1, and TNF-α present in the BAL were reduced in animals treated with naïve and cytomix-pre-activated MSCs compared to the vehicle control (**A**–**C**). Interleukin-18 was significantly increased in all MSC-treated groups compared to the control (**D**). Interleukin-12 was significantly increased in the cytomix and hypoxia MSC groups compared to the control (**E**). MIP-1α was reduced by naïve and cytomix MSCs (**F**), and MIP-3α was significantly reduced by naïve and cytomix-pre-activated MSCs compared to the control (**G**). MMP-9 and MCP-1 were significantly reduced in the naïve and cytomix-pre-activated MSC groups (**H**,**I**). IL = interleukin; CINC-1 = cytokine-induced neutrophil chemoattractant 1; TNF-α = tumor necrosis factor alpha; MIP = macrophage inflammatory protein; MMP = matrix metalloprotease; MCP = monocyte chemotactic protein. Columns represent means. Error bars represent SDs. *, **, *** = *p* ≤ 0.05, 0.01, 0.001 with respect to the vehicle control. $$, $$$, $$$$ = *p* ≤ 0.01, 0.001, 0.0001 with respect to the sham group. # = *p* ≤ 0.05 with respect to other treatment. *N* = 8–12.

**Figure 5 pharmaceuticals-16-00149-f005:**
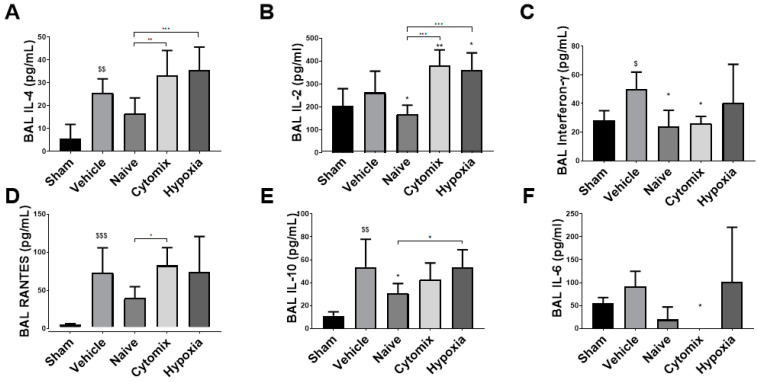
The T cell regulatory cytokines IL-4 and IL-2 were significantly decreased in the BAL with the administration of naïve MSCs and increased with the administration of cytomix- and hypoxia-pre-activated MSCs (**A**,**B**). IFN-γ was significantly reduced by both naïve and cytomix-pre-activated MSCs compared to the vehicle control (**C**). The chemotactic and leukocyte activation regulator RANTES was reduced by naïve MSC treatment compared to the vehicle (**D**), as was anti-inflammatory IL-10 (**E**) and inflammatory marker IL-6 (**F**). IL = interleukin; RANTES = regulated upon activation normal T cell expressed and presumably secreted. Columns represent means. Error bars represent SDs. *, **, *** = *p* ≤ 0.05, 0.01, 0.001 with respect to the vehicle control. $, $$, $$$ = *p* ≤ 0.05, 0.01, 0.001 with respect to the sham group. *N* = 8–12.

**Figure 6 pharmaceuticals-16-00149-f006:**
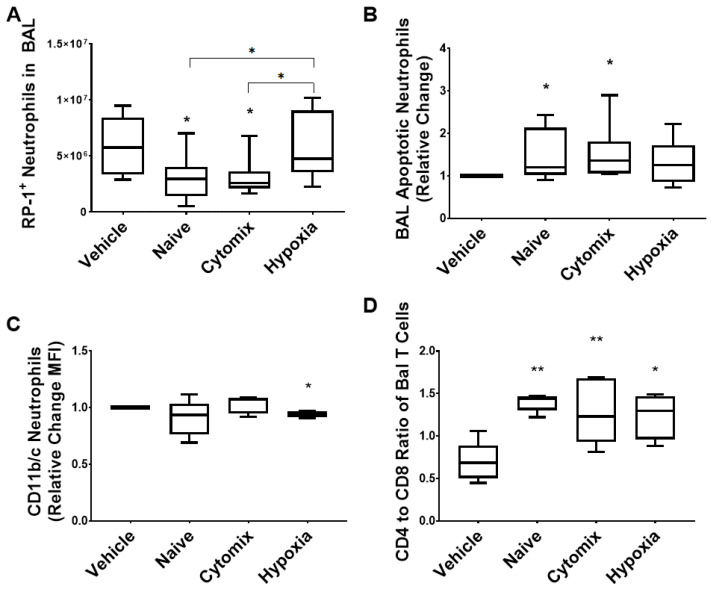
Flow cytometry analysis revealed that the total number of RP-1^+^ neutrophils present in the BAL fluid was significantly reduced in animals treated with naïve and cytomix-pre-activated MSCs compared to the control (**A**). The number of these neutrophils undergoing apoptosis was increased in the BAL fluid of animals treated with naïve or cytomix-pre-activated cells (**B**), and the number of activated neutrophils was slightly reduced by the application of naïve and hypoxia-pre-activated cells (**C**). CD4^+^ T helper cells in the BAL fluid remained largely unchanged across the groups (**D**). BAL = bronchoalveolar lavage. Box plots and whiskers represent the minimum, first quartile, median, third quartile, and maximum. *, ** = *p* ≤ 0.05, 0.01 with respect to the vehicle control. *N* = 8–12.

**Figure 7 pharmaceuticals-16-00149-f007:**
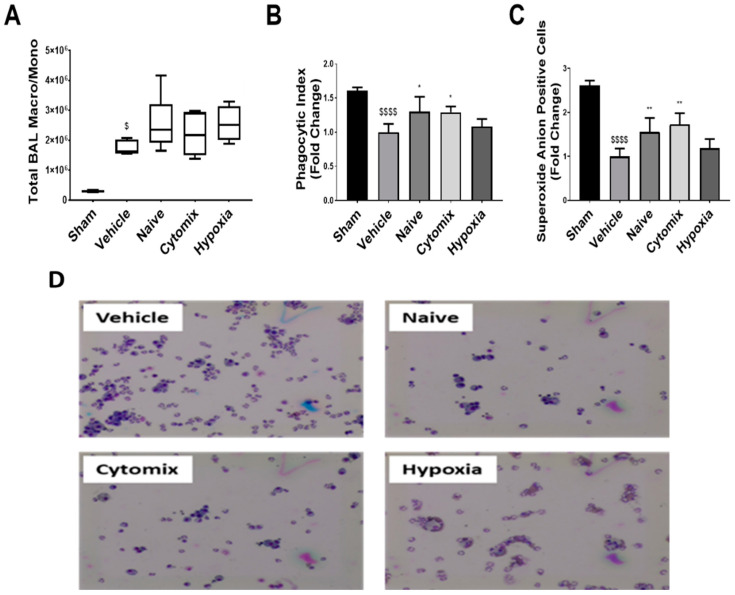
Microscopic analysis of Diff-Quik-stained cytomix BAL cells show an increased infiltration into the alveolar airspace of macrophages/monocytes with the *K. pneumoniae* injury but no change to the total amount between treatment groups (**A**); representative images of treatment groups are shown (**D**). Adherent BAL macrophage/monocyte phagocytosis (**B**) and superoxide anion production (**C**) revealed a significant reduction in cell function after the injury, with naïve and cytomix-pre-activated MSCs significantly increasing their function. BAL = bronchoalveolar lavage. Box plots and whiskers represent the minimum, first quartile, median, third quartile, and maximum *, ** = *p* ≤ 0.05, 0.01 with respect to the vehicle control. $, $$$$ = *p* ≤ 0.05, 0.0001 with respect to the sham group. *N* = 8–12.

**Figure 8 pharmaceuticals-16-00149-f008:**
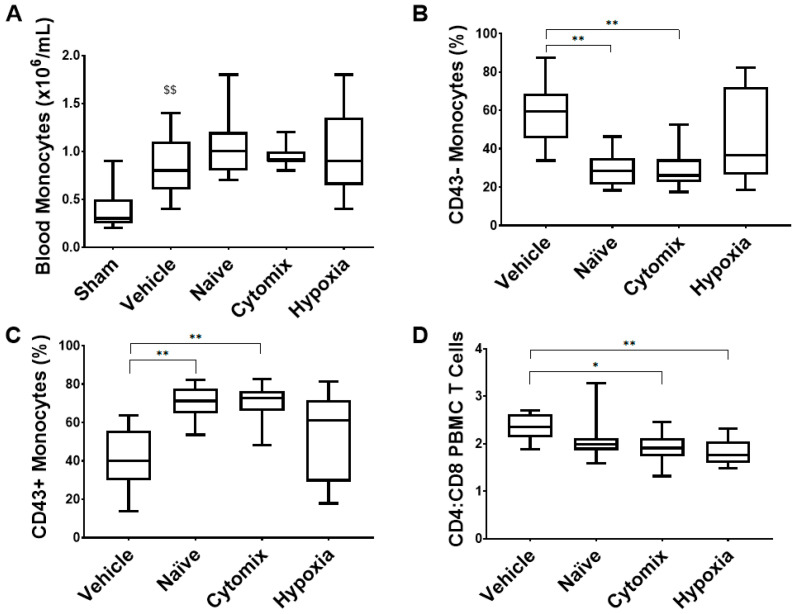
The monocyte fraction increased after treatment with MSCs (**A**), and the % of those that were classical monocytes was decreased by naïve and cytomix MSC treatments (**B**). Nonclassical monocytes were increased in the same groups (**C**). The CD4 to CD8 ratio was significantly decreased with the administration of cytomix- and hypoxia-pretreated MSCs (**D**). PBMC = peripheral blood mononuclear cell; Treg = regulatory T cell. Box plots and whiskers represent the minimum, first quartile, median, third quartile, and maximum. *, ** = *p* ≤ 0.05, 0.01 with respect to the vehicle control. $$ = *p* ≤ 0.01 with respect to the sham group. *N* = 8–12.

**Figure 9 pharmaceuticals-16-00149-f009:**
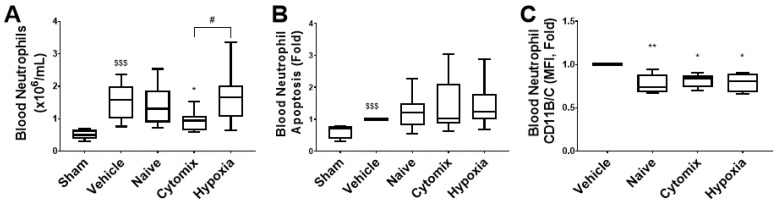
Flow cytometry analysis revealed that the population of circulating neutrophils increased in response to *K. pneumoniae* infection, and this was significantly attenuated by the administration of cytomix MSCs (**A**). The percentage-fold change in neutrophil apoptosis appeared to be increased (**B**), and the activation was significantly decreased by all MSC-treated groups (**C**). MFI = median fluorescence intensity; IL = interleukin; TNF-α = tumor necrosis factor alpha. Box plots and whiskers represent the minimum, first quartile, median, third quartile, and maximum. Columns represent means, and error bars represent SDs. *, ** = *p* ≤ 0.05, 0.01 with respect to the vehicle control. $$$ = *p* ≤ 0.001 with respect to the sham group. # = *p* ≤ 0.05 with respect to other treatment. *N* = 8–12.

## Data Availability

Datasets are available at https://doi.org/10.6084/m9.figshare.21918849.

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
