# Peer review of "Differential Effects of Cytokine Versus Hypoxic Preconditioning of Human Mesenchymal Stromal Cells in Pulmonary Sepsis Induced by Antimicrobial-Resistant Klebsiella pneumoniae"

_pharmaceuticals, 2023, doi:10.3390/ph16020149_

Round 1
Reviewer 1 Report
This work by Byrnes et al contributes to the body of literature demonstrating how ‘priming’ MSCs can enhance the therapeutic properties of these cells. In particular, the authors demonstrate that priming human umbilical-cord derived MSCs with a cytokine mix were more effective than naïve or hypoxia induced MSCs at reducing inflammation and resolving bacterial induced lung injury in animals.
1. It is difficult to interpret Figure 1A because the symbols and error bars overlap. It would help the reader understand this data if the samples were color coded in this plot to differentiate sample conditions. The y-axis appears to depict the % wound remaining, and the words ‘fold change’ should be removed.
2. It is unclear why each assay in Figure 1A-C used a different injury model. Perhaps the authors can provide some rationalization in the text for each injury model.
3. Results section line 108 should specify what route of administration was used.
4. Figure 3 panels A-D are missing labels or descriptions regarding what each panel is depicting. Are these panels depicting representative images of the vehicle, naïve, cytomix and hypoxia conditions quantified in 3E? If so, the airspace quantification in 3E seems inconsistent with the images. For example, it appears there is a dramatic increase in airspace between the microscopy panels A & B, but the quantification data suggests a difference of ~1% between those conditions. Perhaps there is an alternative or higher resolution quantification method that can be used to quantify the images more accurately? Figure legend should also mention the stain used and a scale bar should be included in at least 1 panel for reference.
5. Sham controls are inconsistently included throughout the manuscript, even between panels within the same figure. Can the authors include the sham control data for all experiments if available, or address why they are excluded from certain panels?
6. This manuscript includes non-published data that is listed as “not shown” in the text. It appears that these data are excluded because they do not demonstrate a statistically significant difference. However, I believe this data is still informative for this study and should be included. Perhaps these plots can be included in a supplementary section.
7. There is a section in the methods section about “BAL macrophage/monocyte phagocytosis and superoxide anion production”, as well as a sentence in the discussion starting on line 357 referring to results from these experiments. However, these experiments appear to be absent from this manuscript. Were these experiments removed from the study, or added to a different version?
8. There are experiments in Figure 1 involving the antibacterial properties of MSCs and how priming MSCs can enhance those traits. It is interesting that the hypoxia primed MSCs were more effective at directly inhibiting bacterial growth in vitro, but were less effective in the in vivo models. A sentence or two with references describing what is known about the antibacterial properties of MSCs and how priming impacts those traits should be included somewhere in the intro or discussion.
9. Did the authors test whether the systemically administered MSCs arrived in the lung in these experiments? Some previous studies have used inhalation or intranasal administration of MSCs for a more direct route of exposure to lung injury. Do the authors have a hypothesis on how the route of administration may impact the therapeutic properties tested in this study? And finally, it could be possible that the cytomix and hypoxia primed MSCs have differential trafficking to the lung. Perhaps the cytomix primed cells appear more effective in this model because they are more effective at trafficking to sites of infection in the lung. These experiments are beyond the scope of this study, but some discussion with relevant references should be added to the discussion to touch on some of these points.
Author Response
General Comment: This work by Byrnes et al contributes to the body of literature demonstrating how ‘priming’ MSCs can enhance the therapeutic properties of these cells. In particular, the authors demonstrate that priming human umbilical-cord derived MSCs with a cytokine mix were more effective than naïve or hypoxia induced MSCs at reducing inflammation and resolving bacterial induced lung injury in animals.
Response: The authors would like to thank the reviewer for their thorough and comprehensive review of our manuscript. We have no doubt that the changes suggested and implemented will enhance the quality of our manuscript. Please see our individual responses below to each of your comments and suggestions. We trust that these amendments will be satisfactory.
Comment 1: It is difficult to interpret Figure 1A because the symbols and error bars overlap. It would help the reader understand this data if the samples were color coded in this plot to differentiate sample conditions. The y-axis appears to depict the % wound remaining, and the words ‘fold change’ should be removed.
Response: The authors would like to thank the reviewer for alerting us to this.
Figure 1A has been revised to include colour coding and enlarged to aid in interpretation. As such the layout of the figure has been changed but other graphs remain the same. Fold change has been removed from the Y-axis.
Comment 2: It is unclear why each assay in Figure 1A-C used a different injury model. Perhaps the authors can provide some rationalization in the text for each injury model.
Response: Thank you for this query. During our in vitro testing, we attempted to model in vitro the different injury and repair processes which occur in the lung during infection. Hence, we have a tissue repair following injury (wound assay), induction of inflammation (NFκB assay), and the production of reactive oxygen species model (metabolic activity) which are all common injurious pathways observed. Additional text has been added to the manuscript to clarify this:
Line 86-88 ‘By simulating the different injury and repair processes seen in the lung over the course of a pneumonia injury, it was hoped to model the direct effects of our MSC therapy in vitro.’,
Line 302-303 ‘which is why wound-closure, NF-kappa-B activation, and ROS injury assays were included.’
Comment 3: Results section line 108 should specify what route of administration was used.
Response: We wish to thank the reviewer for pointing out this oversight. The sentence has been amended as follows:
Line 122, “The intravenous (IV) administration of cytomix pre-activated MSCs via tail-vein to rodent models…”
Comment 4: Figure 3 panels A-D are missing labels or descriptions regarding what each panel is depicting. Are these panels depicting representative images of the vehicle, naïve, cytomix and hypoxia conditions quantified in 3E? If so, the airspace quantification in 3E seems inconsistent with the images. For example, it appears there is a dramatic increase in airspace between the microscopy panels A & B, but the quantification data suggests a difference of ~1% between those conditions. Perhaps there is an alternative or higher resolution quantification method that can be used to quantify the images more accurately? Figure legend should also mention the stain used and a scale bar should be included in at least 1 panel for reference.
Response: Thank you for your thorough examination of figure 3. The figure and figure legend have been amended as follows:
Figure 3 was amended to include the sham group and the format was changed from a bar graph to box and whiskers for better representative error bars to show the increased spread of data from the vehicle group.
Representative images were changed to show large white space even in collapsed groups. Since the lobe is divided into 5 sections, with an average of three slices taken from each section, there is a lot of data points for the N=8-12, and so the power of the statistical analysis is high. The mean percentage airspace in vehicle treated animals was 76.45% while cytomix was 79.3%, median was 78.72% and 79.46 respectively. In contrast the sham animals presented a mean airspace percentage of 82.81% demonstrating that there was a significant injury.
The figure legend was changed to include:
Line 155 ‘H&E stained’,
Line 156-157 ‘Representative images of sham (B), vehicle (C), naïve (D), cytomix (E), and hypoxia (F) histology.’
Comment 5: Sham controls are inconsistently included throughout the manuscript, even between panels within the same figure. Can the authors include the sham control data for all experiments if available, or address why they are excluded from certain panels?
Response: Thank you for the opportunity to clarify this, your comments have been addressed as detailed below.
Sham data has now been included in Figure 3 which was an oversight on our part, we apologise for this.
The abundance of neutrophils seen in our infection model is not evident in sham controls. We did not have sufficient numbers to conduct experiments in healthy animals and this has been clarified in the text as follows:
Line 213-214 ‘Sham data was not included due to the sparsity of neutrophils in the BAL of healthy rodents.’ which is also the case for supplementary figure 3.
Line 258-259 ‘Sham monocyte populations were not assessed due to limited sampling from healthy controls.’.
Line 272-273 ‘Sham CD11B/C median fluorescent intensity (MFI) was not included as could not per-form relative fold change for that sample.’ as this was relative fold change it required a sham at every harvest to compare to vehicle which was not possible (all shams were from the same batch of animals) which is also the case for supplementary figure 4
Comment 6: This manuscript includes non-published data that is listed as “not shown” in the text. It appears that these data are excluded because they do not demonstrate a statistically significant difference. However, I believe this data is still informative for this study and should be included. Perhaps these plots can be included in a supplementary section.
Response: We thank you for this comment and agree that this data is informative for this study. To address this, a supplementary section has been created to include any of the ‘not shown’ data along with the associated materials and methods. Appropriate changes have been made to the manuscript to represent this.
Other relevant material has also been included to aid in explaining some of the reviewers’ comments which include a Supplementary Figure 1 which has in vitro data associated with the data in text with Figure S1B correlates with what is seen in vivo. Supplementary Figure 2 shows the characterisation of the MSC surface markers.
Comment 7: There is a section in the methods section about “BAL macrophage/monocyte phagocytosis and superoxide anion production”, as well as a sentence in the discussion starting on line 357 referring to results from these experiments. However, these experiments appear to be absent from this manuscript. Were these experiments removed from the study, or added to a different version?
Response: Apologies, this has now been rectified. We had excluded the figure in a post submission draft. The figure has now been included here for your review as follows:
Section 2.6 has been added at line 226-240, Figure 7 has been added at line 242 with the appropriate figure legend.
An additional materials and method section 4.4.7 has been added due to one of the included graphs (Figure 7A) and the associated images. All Figure numbers were adjusted in text.
Comment 8: There are experiments in Figure 1 involving the antibacterial properties of MSCs and how priming MSCs can enhance those traits. It is interesting that the hypoxia primed MSCs were more effective at directly inhibiting bacterial growth in vitro, but were less effective in the in vivo models. A sentence or two with references describing what is known about the antibacterial properties of MSCs and how priming impacts those traits should be included somewhere in the intro or discussion.
Response: Thank you for this helpful comment. Additional text has been added to the introduction section to address this. Information about MSC anti-microbial properties has been added to line 40 – 45: ‘Mesenchymal stromal cells (MSCs) have previously been shown to have potent anti-bacterial properties [4] [5] along with anti-viral [6] and anti-parasitic [7] effects, both directly affecting microbial infections and indirectly through their immunomodulatory properties. Directly, the MSCs release anti-microbial peptides and this can be enhanced through pre-conditioning the cells with bacterial compounds and inflammatory cytokines [8].’
Comment 9: Did the authors test whether the systemically administered MSCs arrived in the lung in these experiments? Some previous studies have used inhalation or intranasal administration of MSCs for a more direct route of exposure to lung injury. Do the authors have a hypothesis on how the route of administration may impact the therapeutic properties tested in this study? And finally, it could be possible that the cytomix and hypoxia primed MSCs have differential trafficking to the lung. Perhaps the cytomix primed cells appear more effective in this model because they are more effective at trafficking to sites of infection in the lung. These experiments are beyond the scope of this study, but some discussion with relevant references should be added to the discussion to touch on some of these points.
Response: The authors welcome the chance to address this point. While in the context of these experiments we deemed the route of administration to be outside of our focus, never-the-less it is a crucial point to consider.
A past publication from our group looked directly at the administration of MSCs to mouse models of pneumonia alongside healthy controls. The MSCs were shown in both models to accumulate almost immediately in the pulmonary vasculature when given either by tail vein or jugular vein. The longevity of these cells in the lungs differed between the healthy and pneumonia animals where cells were not present in any significant number 24h after administration in healthy models, but persisted in the pneumonia animals in high numbers. This was tested in vitro using LPS primed endothelial layers and found to be due to the activation status of the vascular cells. Separately, LPS and TNF-a primed cells were shown to adhere better to pulmonary endothelial cells in a static adhesion model in vitro but this was not taken to in vivo models due to time constraints.
A sentence has now been to the introduction and discussion sections of this manuscript to highlight our background work and that this was a consideration which should be included in future studies.
Line 70-74: It has been shown that MSCs home to sites of injury and inflammation, and our previous studies have demonstrated that regardless of the inflammation state, IV-administered MSCs become lodged in the lung vasculature im-mediately after administration. The MSCs are cleared within 24h in healthy lungs but remain there for over 24h in states of active infection.
Line 455-458: Future studies should build on these results and address other factors such as route of administration, the optimal MSC or MSC product, and different methods of cell preservation to enable a better understanding of the mechanisms of action of MSC therapy.
Reviewer 2 Report
In the present work, Byrnes et al comparatively analyze MSCs pre-stimulated with cytokines versus those expanded in hypoxia both in vitro and in an animal model of Klebsiella pneumoniae pulmonary sepsis.
The work continues a relevant line of research of this group, and contributes to a better understanding of the role of MSCs in this context.
To improve the manuscript I would like to make the following comments:
- In the methodology section, there is a complete section on the analysis of phagocytosis by monocytes/macrophages and superoxide anion production but there is no information about it in the results or in the supplementary data. This information should obviously appear in the results and these should be adequately discussed.
- The main data of the paper, that MSCs in hypoxia do not have the same function in vivo as pre-stimulated MSCs (despite the in vitro data, where it even seems that MSCs in hypoxia do better) should be further discussed and elaborated with the state of the literature in this field, as it is a controversial topic and these data are of much interest.
- The opposite effects on CD43+ and CD43-negative monocytes should be interpreted in the discussion, and the information should be expanded a bit for proper understanding.
-Some typographical errors should be corrected (e.g. 2x105 BAL cells, line 521), with2mg/mL, line 527).
Author Response
General Comment: In the present work, Byrnes et al comparatively analyze MSCs pre-stimulated with cytokines versus those expanded in hypoxia both in vitro and in an animal model of Klebsiella pneumoniae pulmonary sepsis.
The work continues a relevant line of research of this group, and contributes to a better understanding of the role of MSCs in this context.
Response: The authors would like to thank the reviewer for their assessment of our manuscript and their constructive and welcome comments.
Comment 1: In the methodology section, there is a complete section on the analysis of phagocytosis by monocytes/macrophages and superoxide anion production but there is no information about it in the results or in the supplementary data. This information should obviously appear in the results and these should be adequately discussed.
Response: We thank the reviewer for this observation - this has now been rectified. Section 2.6 has been added at line 226, Figure 7 has been added at line 242 with the appropriate figure legend. An additional materials and method section 4.4.7 has been added due to one of the included graphs and associated images. Relevant figure numbers have been adjusted in text.
Comment 2: The main data of the paper, that MSCs in hypoxia do not have the same function in vivo as pre-stimulated MSCs (despite the in vitro data, where it even seems that MSCs in hypoxia do better) should be further discussed and elaborated with the state of the literature in this field, as it is a controversial topic and these data are of much interest.
Response: We thank the reviewer for these comments and agree – the data are of much interest! A supplementary section has been included to better explain this along with the associated changes in the text. Supplementary Figure 1 has in vitro data which nicely associates with the data in text while Figure S1B correlates with what is seen in vivo. Supplementary Figure 2 shows the characterisation of the MSC surface markers.
Line 98: ‘(Figure 1B) and LPS injury (Figure S1A)’,
Line 99-100: ‘with hypoxia pre-activation only reducing TNF-α secretion on average (Figure S1A). All MSC-CM’,
Line 102-104: ‘Cytomix pre-activated MSC-CM enhanced neutrophil-like expression of CD11b while hypoxia MSC-CM significantly reduced their expression (Figure S1B).’.
Line 124-126: ‘MSCs were characterised according to ISCT surface marker expression (Figure S2A-C) showing a reduced CD105 expression for hypoxia MSC while cytomix had an in-creased HLA-DR expression.’.
Line 295-299: ‘Characterisation of the cells post-cryopreservation showed a large population of CD105- MSCs after hypoxia pre-activation. This population of cells has been associated with enhanced immunosuppression [25] as shown by the reduced CD11b activation on neutrophil-like cells yet they were less effective in modulating the macro-phage-like inflammatory response.’
A section in the discussion at line 424-437 has been added to aid in the explanation on the ineffectiveness of hypoxia in our model, we believe these cells may be better suited in auto-immune diseases.
Comment 3: The opposite effects on CD43+ and CD43-negative monocytes should be interpreted in the discussion, and the information should be expanded a bit for proper understanding.
Response: We welcome this suggestion and additional text has been added to the discussion section as follows:
Line 392-408: ‘Monocytes are antigen presenting cells (APCs) and part of the professional phagocytes within the innate immune system. They play an important role in maintaining tissue and immune homeostasis [52]. Classical monocytes have enhanced phagocytic capacity and are primed for migration and immune cell signalling [53]. Non-classical monocytes patrol the endothelium in search of injury, antigen presentation to T-cells, and partake in complement and Fc gamma-mediated phagocytosis and adhesion [53]. Rat monocytes subpopulations can be identified based on their level of CD43 expression [54]. Monocytes have increased survival, phagocytosis, activation, APC potential, and adapt a regulatory phenotype in the presence of MSCs [55][56][57]. Previous literature describes MSC's ability to reprogram inflammatory classical to anti-inflammatory non-classical monocytes leading to overall therapeutical benefits in animal models in part mediated by the suppression of T-cell proliferation [58]. Here we demonstrated that both naïve and cytomix pre-activated MSC treatment significantly in-creased the proportion of non-classical monocytes to classical in circulation which may partially explain the improved lung function and structure after cytomix pre-activation. However, further research is needed into their activation state and function after MSC treatment as this can drastically alter their role [59].’.
Comment 4: Some typographical errors should be corrected (e.g. 2x105 BAL cells, line 521), with2mg/mL, line 527).
Response: The authors thank the reviewer for these observations, both of these changes have been implemented.